Male emergence schedule and dispersal behaviour are modified by mate availability in heterogeneous landscapes: evidence from the orange-tip butterfly

Davies W James wjdavies@liv.ac.uk
Saccheri Ilik J.
Institute of Integrative Biology, University of Liverpool , Liverpool , UK
Yoccoz Nigel
Electronic publication date: 2015 Jan 6
Publication date: 2015
Volume: 3
Electronic Location ID: e707
Received 2014 Aug 6; Accepted 2014 Dec 1
Copyright: © 2015 Davies and Saccheri
Copyright year: 2015
Copyright holder: Davies and Saccheri
License: This is an open access article distributed under the terms of the Creative Commons Attribution License, which permits unrestricted use, distribution, reproduction and adaptation in any medium and for any purpose provided that it is properly attributed. For attribution, the original author(s), title, publication source (PeerJ) and either DOI or URL of the article must be cited.
License URL: https://creativecommons.org/licenses/by/4.0/

Keywords: Eclosion timing, Protandry, Dispersal polymorphism, Informed dispersal, Source-sink populations, Landscape structure, Anthocharis cardamines

Funding: This research had no external funding.

==============================
Protandry (prior emergence of males) in insect populations is usually considered to be the result of natural selection acting directly on eclosion timing. When females are monandrous (mate once), males in high density populations benefit from early emergence in the intense scramble competition for mates. In low density populations, however, scramble competition is reduced or absent, and theoretical models predict that protandry will be less favoured. This raises the question of how males behave in heterogeneous landscapes characterized by high density core populations in a low density continuum. We hypothesized that disadvantaged late emerging males in a core population would disperse to the continuum to find mates. We tested this idea using the protandrous, monandrous, pierid butterfly Anthocharis cardamines (the orange-tip) in a core population in Cheshire, northwest England. Over a six-year period, predicted male fitness (the number of matings a male can expect during his residence time, determined by the daily ratio of virgin females to competing males) consistently declined to <1 in late season. This decline affected a large proportion (∼44%) of males in the population and was strongly associated with decreased male recapture-rates, which we attribute to dispersal to the surrounding continuum. In contrast, reanalysis of mark-release-recapture data from an isolated population in Durham, northeast England, showed that in the absence of a continuum very few males (∼3%) emerged when fitness declined to <1 in late season. Hence the existence of a low density continuum may lead to the evolution of plastic dispersal behaviour in high density core populations, maintaining late emerging males which would otherwise be eliminated by selection. This has important theoretical consequences, since a truncated male emergence curve is a key prediction in game theoretic models of emergence timing which has so far received limited support. Our results have implications for conservation, since plastic dispersal behaviour in response to imperfect emergence timing in core (source) populations could help to maintain sink populations in heterogeneous landscapes which would otherwise be driven to extinction by low mate encounter-rates (Allee effects).

Introduction

Protandry, in its broadest sense, denotes the input of males before females into breeding areas (Morbey & Ydenberg, 2001). In insects, where mating usually takes place close to the eclosion (emergence) site, protandry refers more specifically to the prior emergence of males (Wiklund & Fagerstrom, 1977). This widespread phenomenon has repeatedly been analyzed in terms of direct selection pressures acting on emergence timing (Wiklund & Fagerstrom, 1977; Botterweg, 1982; Fagerstrom & Wiklund, 1982; Bulmer, 1983; Iwasa et al., 1983; Parker & Courtney, 1983; Zonneveld & Metz, 1991; Iwasa & Haccou, 1994). Although incidental explanations based on independent selection for correlated traits are also possible (Morbey & Ydenberg, 2001), these are considered unlikely in insects (e.g., Wiklund & Solbreck, 1982; Wiklund, Wickman & Nylin, 1992; Holzapfel & Bradshaw, 2002; but see Matsuura, 2006).

The key requirements for the evolution of protandry in insects are that generations are discrete, so that prior male emergence is possible (Singer, 1982), and that females mate only once (monandry), so that late emergence is costly to males (Wiklund & Fagerstrom, 1977). The earliest mathematical attempts to explain protandry were optimality models based on the assumption that maximum reproductive success coincides with peak emergence in males (Wiklund & Fagerstrom, 1977) or with peak male availability in females (Fagerstrom & Wiklund, 1982). Thus, only the average date of emergence was taken to be under selective control; the variance around this date was considered to be caused by environmental noise (Fig. 1A). Reproductive success in males was measured by the number of matings expected over their lifetime (based on the relative numbers of virgin females to male competitors through the season) or in females by the rapidity with which they were mated (based on the number of males in the population at the time of their eclosion). These models worked well in so far as an earlier emergence date was predicted for males, whether males were considered to be selected in response to the female emergence curve (Wiklund & Fagerstrom, 1977) or vice versa (Fagerstrom & Wiklund, 1982). An alternative model, based on the equilibration of the male emergence curve at the point where directional selection on its mean ceases, was also successful in predicting protandry (Bulmer, 1983).

Figure 1 Comparison of the key features of different protandry models. In all cases, selection modifies male emergence date (filled symbols) in response to the female emergence curve (open symbols).

Abscissa, emergence date (arbitrary units); ordinate, number emerging. (A) Protandry results from selection on peak male emergence date (vertical line); (B) frequency-dependent selection modifies the shape, as well as the mean, of the male emergence curve, which is predicted to be truncated; (C) a bet-hedging strategy increases the variance of the male emergence curve in response to stochasticity in female emergence date (here represented by varying position of female emergence curve); (D) mate encounter rate modifies degree of protandry, which is more pronounced in high (circles) than in low (squares) density populations (ordinate, number per unit area (density) emerging; this model reverts to the assumption that only the mean emergence date is modified by selection).

The validity of the assumption that only the average emergence date is modified by selection was challenged in several game-theoretic models in which it is replaced by the hypothesis that specimens emerging at any point in the season gain equal fitness (Bulmer, 1983; Iwasa et al., 1983; Parker & Courtney, 1983). This situation is assumed to evolve in response to frequency-dependent selection: males emerging in peak season will encounter the highest number of females but will also be in competition against the highest number of males, whereas those emerging at other times will encounter fewer females but will also have fewer competitors. In order to balance the fitness of specimens emerging at any point in the season exactly, the male emergence curve must take a specific form in relation to the female emergence curve. Hence, both the peak and the shape of the male emergence curve are envisioned as responding to selection, which pushes the emergence schedule towards an ideal free distribution or evolutionarily stable strategy (ESS). If female emergence timing is treated as an independent variable, then the corresponding ESS male emergence schedule can be solved for it either analytically or by simulation. (Game theorists have so far paid little attention to how the female emergence schedule might respond to male emergence timing (but see Zonneveld & Metz, 1991)).

The results of the early game-theoretic models were mixed. A key prediction of the analytic models was that the male emergence curve should be truncated (Bulmer, 1983; Iwasa et al., 1983), with no males emerging after a specific date in the season (Fig. 1B); this was not observed in careful studies of the checkerspot butterfly Euphydryas editha (Iwasa et al., 1983; Baughman, Murphy & Ehrlich, 1988). Iwasa & Haccou (1994) suspected that deterministic game-theoretic models, which neglected stochastic noise, relied too heavily on the implicit assumption that organisms possess extremely accurate emergence cues which would enable them to perfectly compensate for the effects of a fluctuating environment. This led them to examine the impact of stochastic effects on the male emergence curve in relation to a bet-hedging strategy, in which specimens of the same genotype emerge at different times to maximize their average logarithmic reproductive success. They found that, in the absence of an accurate cue, such a strategy increased the variance in the emergence curve compared with that predicted by a deterministic model (Fig. 1C); in the presence of a perfect cue, the emergence schedule is identical to the one predicted by the deterministic model. This work was important in stressing the relevance of environmental noise and emergence cues in the evolution of protandry (Sawada et al., 1997), but the problems relating to the prediction of a truncated emergence remained.

The simulation model of Parker & Courtney (1983) fared rather better when applied to the orange-tip butterfly, Anthocharis cardamines. In a highly localized population in Durham in northeast England, the observed distribution of male eclosion times closely matched the predicted ESS distribution (whether calculated on the assumption that females were (partly) polyandrous or monandrous). Interestingly, the male emergence curve for this population did terminate abruptly (neglecting the contribution of a small number of late emerging specimens, and within the limits set by the summation of emergences over successive 4-day periods). These results indicate that strong selection is capable of effectively modifying male emergence timing in this species.

A key assumption in all the protandry models discussed so far is that females are mated on the day of eclosion, so the presence curve for virgins is identical to the emergence curve. In low density populations, where mate encounter rates are low, this is unlikely to be true. Theoretical evidence that protandry should evolve in response to population density was provided by Zonneveld & Metz (1991). Their analysis reverted to the assumption that only mean emergence time is under selective control, and they used a ‘law of mass action’ (density-dependence) to model male–female encounter rates. They found that as encounter rates approach zero, the evolutionarily stable degree of protandry also approaches zero; hence, protandry should be diminished or absent in low density populations (Fig. 1D). These results are in agreement with those of the earlier simulation model of Botterweg (1982) for the pine looper moth Bupalus piniarius, which showed that the expected degree of protandry decreases when both flight activity (males) and moth density (both sexes) are reduced to very low levels, i.e., when mate encounter rates are minimized.

In heterogeneous landscapes, population density will vary spatially, so protandry will be more favoured in some areas than in others. Specifically, protandry should be strongly selected in high density core habitats; if these are isolated or nearly so, and if selection is frequency-dependent, late emerging males should be eliminated, since a truncated emergence curve is a robust prediction of the analytic game-theoretic models (Iwasa & Haccou, 1994). If, however, the core habitats are not isolated, but connected to low density areas in which protandry is less favoured, late emerging males could increase their fitness (i.e., the number of matings they can expect) by emigrating to them (Fig. 2).

Figure 2 Schematic representation of hypothesized male behaviour in a high density core population immersed in a low density continuum.

Models (B) and (D) in Fig. 1 predict that a truncated male emergence curve should evolve in response to frequency-dependent selection and that protandry should be more pronounced in high density populations. In heterogeneous landscapes, late emerging core males could therefore improve their fitness by emigrating to low density areas; selection for such behaviour could prevent the evolution of a truncated emergence curve. Filled and open circles show male and female emergence curves; grey and black shading represent adaptive and maladaptive male emergence timing in isolated populations, with the predicted truncation date lying at the boundary between them; the arrow shows the fitness benefit gained by late emerging core males in dispersing to the continuum; abscissa, arbitrary emergence date; ordinate, relative density emerging.

In this paper, we test the key predictions of this hypothesis for a species (A. cardamines) which has been shown to exhibit a truncated emergence curve in an isolated population (Parker & Courtney, 1983). In a high density core population immersed in a low density continuum we investigate the following questions:

1. Is there a consistent decline in predicted male fitness through the flight season, indicating that if the core habitat were isolated, late emerging males would be eliminated (as in the population studied by Parker and Courtney)?

2. Do disadvantaged late season core males emigrate to the continuum?

Methods

Study organism and study site

A detailed description of the study site (including a map) and the field work undertaken there is given in Davies & Saccheri (2013). Briefly, Dibbinsdale Nature Reserve encompasses a large (475 ha) remnant of semi-natural ancient woodland on the Wirral peninsula in Cheshire, northwest England. A. cardamines is a univoltine springtime butterfly of damp meadows, riverbanks, open woodland, hedgerows and lanes which emerges in April and May; females are usually monandrous (Wiklund & Forsberg, 1991). Males in compact populations (including Dibbinsdale) locate females by patrolling behaviour (repeatedly flying back and forth over the same ground). The two host-plants most commonly utilized in Britain (Cardamine pratensis and Alliaria petiolata) are both abundant in Dibbinsdale, which we regard as a ‘core’ habitat for A. cardamines; A. petiolata is sporadically distributed along road margins in a wide area outside the Reserve, which we term the ‘continuum.’ (The continuum extends much further than the area shown in the map in Davies & Saccheri (2013); its utilization by A. cardamines has been confirmed by the occurrence of larvae there). Field work (mark-release-recapture) was undertaken every day of the flight period. The butterflies were individually marked, and released where captured. None were observed to undertake an escape flight. The dates of all recaptures were recorded. The data for each year in the study period (2005–2010) were analyzed separately. Poor weather days were excluded from the analyses, since their inclusion has been found to markedly disrupt mark-release-recapture data.

POPAN estimation of male and female emergence schedules

The daily ‘input’ (emergence and immigration) of males and females into the Dibbinsdale population in each year of the study period was estimated using the POPAN formulation in the program MARK (Cooch & White, 2014). The usual approach to analyzing mark-release-recapture (MRR) data with MARK is to start with a general model and then modify it through parameter reduction. This results in a set of candidate models from which hypotheses relating to the behaviour of the marked animals can be tested. Since the number of sampling occasions in our data-sets were large (25–46 days), the most general time-dependent models which could have been constructed would have been extremely unwieldy and of very limited utility due to the sparseness of encounters outside peak season. Moreover, we already possessed accurate information on the behaviour of the butterflies from analyses of recapture-duration decay-plots (Davies & Saccheri, 2013). We therefore built our models directly on the basis of this information (after confirming there were no inherent problems with our data by assessing goodness-of-fit with the Release tool in MARK). Different approaches were used for males and females.

Male behaviour is influenced by both size and time of appearance. Individuals were therefore assigned to separate attribute groups on the basis of wing-length and intra-seasonal period of first capture. Each group was assigned a separate constant survival probability (residence-rate) and the encounter probability was set to the same constant in all groups (as sampling effort was uniform). The total number of new entrants into the population on each day of the season was obtained by summing the contributions from each group.

We have no evidence that size or time of appearance affects the behaviour of females. However, there was a problem with unequal catchabilities between different areas of the Reserve in 2005 and 2006, when females were more commonly encountered in some areas than in others. These were accordingly assigned to separate attribute groups with distinct encounter probabilities; the survival probability was set to the same constant in the two groups. In 2007–2010, the problem of unequal catchabilities did not arise, so there was just one group with a single survival and encounter probability.

The daily input curves are difficult to interpret visually. We therefore present simplified curves in which the input of males and females into the population in successive 4-d periods have been combined. In the male fitness models, however, daily inputs were used except in the case of our reanalysis of the data from Parker & Courtney (1983), in which we used the 4-d input intervals and 4-d residence-rate given by the authors.

Male fitness model

For the calculation of predicted male fitness, we exclude the possibility of increased emigration in late season, since this is hypothesized to evolve in response to the situation we are trying to uncover. We therefore assume that male residence time is constant (equivalent to assuming a constant daily residence-rate—see below), and use the fitness model of Parker & Courtney (1983) to test the null hypothesis that observed differences in emergence timing do not affect the number of matings males obtain in the Dibbinsdale population. The Parker-Courtney model assumes that on any day of the flight season the number of matings a male achieves is directly proportional to the number of newly emerging (virginal) females and inversely proportional to the number of male competitors. Hence, females are assumed to be mated on the day of their emergence, and male fitness is a direct consequence of scramble competition for mates. The number of males present on day t(mt) is calculated from (1) mt=∑n=1n=tMnst−n

where Mn is the number of males entering the population on day n (from POPAN), and s is their daily residence-rate (derived from whole-season residence plots—see below); the factor st−n corrects for the loss of specimens between days n and t. The fitness (λ) of males entering the population on day j is therefore (2) λj=∑n=jn=TFnmn+1sn−j

where Fn is the number of females entering the population on day n (from POPAN) and T is the termination date of the season; the factor sn−j is the probability that a specimen entering the population on day j is still present n−j days later. (We have added 1 specimen to the denominator of the Parker-Courtney model to prevent it from tending to zero at times when the population is very sparse.) For each year in the study period, the total number of females entering the population was adjusted to equal the number of males, since in captivity the sex-ratio is equal (an equal sex-ratio in the field is generally supported by the approximate estimations obtained from POPAN). All calculations were executed on a spreadsheet in Excel.

The average fitness (λav) of males entering the population during a specific intra-seasonal period was obtained from (3) λav=∑j=yj=zMj∑n=jn=TFnmnsn−j∑j=yj=zMj

where the intra-seasonal period runs from days j = y to j = z. We have here dropped the addition of 1 to the denominator of the fitness term since mn → 0 implies Mjsn−j → 0, and so the term vanishes from the summation.

There are three caveats which must be taken account of when considering the applicability of this model.

1. Not all females entering the population will be virginal, due to the immigration of mated individuals from outside the study area. However, provided the immigrants do not alter the pattern of the true emergence curve (the temporal variation in the relative number of virgin females entering the population), this effect may be neglected. Since the Dibbinsdale population is highly concentrated (see map in Davies & Saccheri, 2013), it is unlikely that the immigrant flux will be large enough to swamp the true emergence curve (it is certainly insufficient to obscure differences in wing-length distribution between sub-sites). The same considerations apply to the male emergence curve. Hereafter, we refer to all input curves as emergence curves, neglecting the contribution of immigrants.

2. Females have been assumed to be monandrous, so that only virgins are available for mating. This is largely correct for A. cardamines, although polyandrous females are known (Wiklund & Forsberg, 1991). Again, these latter are unlikely to alter the broad pattern in the temporal availability of receptive females.

3. The model implicitly assumes a homogeneous environment in which the emergence curves are uninfluenced by micro-climatic variation. Ideally, we would have liked to study separate eclosion patterns in different sub-sites within the Reserve, but the scarcity of female captures was prohibitive. However, serious problems would only arise if emergence timing was highly asynchronous between sub-sites, which is unlikely. Furthermore, we restrict our key conclusions to inferences drawn from large changes in average male fitness between lengthy intra-seasonal periods each year; such coarse-scale effects should be unaffected by asynchrony between sub-sites, or by daily sampling artefacts, as is evidenced by their repeatability between years.

Reanalysis of data from Parker & Courtney (1983)

We compare our results, obtained for a high density core population located within a low density continuum, with those obtained by Parker & Courtney (1983) for an almost completely isolated population of A. cardamines in Durham in 1977. For the male emergence curve, we used the data given explicitly by Courtney (1980); for the female emergence curve, we measured the input from Fig. 2 of Parker & Courtney (1983). Parker and Courtney did not exclude poor weather days, and their emergence curves are summed over 4-d intervals. Accordingly, these curves were used in conjunction with the 4-d residence-rate (given by Parker & Courtney, 1983) to estimate male fitness at 4-d intervals from Eq. (2).

Intra-seasonal changes in male behaviour

The prediction that late season males emigrate from the core population was tested indirectly. We first develop a novel method for identifying two co-occurring behavioural phenotypes in a wild insect population, when one of the phenotypes has a very short residence time. These phenotypes must differ in either their death or emigration-rates. We then apply this method to our data and argue that the co-occurrence of two phenotypes in late season most likely represents a dispersal polymorphism.

Theory

The daily probability that an animal will be retained in a population is described by its residence-rate (Watt et al., 1977), which is influenced by both survival and movement; the higher the residence-rate, the longer the residence time. In insect populations, the residence-rate is usually constant with age (at least up to the time of senescence). In this case, the number of individuals remaining against time elapsed since first capture will decline exponentially; logarithmic transformation will then yield a straight line residence plot (termed a recapture-duration decay-plot by Watt et al., 1977) whose gradient is determined by the residence-rate. Hence the residence-rate can be obtained from the gradient of a residence plot for MRR data. This will represent the average phenotype in a population, determined by the average vulnerability to death and the average propensity to emigrate. If the residence-rate changes, then the average death-rate and/or emigration-rate will have changed.

Strictly speaking, the residence-rate obtained from a residence plot only applies to specimens recaptured at least once, since it is derived from the time elapsed between first and last capture. Hence, specimens disappearing from the population very rapidly after first capture do not influence the calculated residence-rate. This leaves open the possibility that a distinct phenotype with a very high death/emigration rate might exist in the population which does not impact on the residence plot, since so few recaptures of it are actually made.

If the fraction of specimens recaptured declines through the season, there are two possibilities (Fig. 3).

(A) The residence-rate of the average phenotype has decreased, due to an increase in either the death or emigration-rate across the entire population. In this case, fewer recaptures are made due to the shorter residence time of the average phenotype.

(B) A second phenotype with very high death/emigration rate has appeared in the population alongside the average phenotype. In this case, fewer recaptures are made due to the rapid disappearance of the new phenotype.

To distinguish these possibilities, the residence-rate specific to the period when recaptures are low should be calculated, which allows estimation of the corresponding predicted number of recaptures (see below and Appendix SI). If this does not differ significantly from the observed number, then the death/emigration rate of the average phenotype in the population is sufficient to explain the low number of observed recaptures. If, however, the observed number of recaptures is significantly lower than the expected number, then the average phenotype is accompanied by a second phenotype with a very high death or emigration-rate.

Figure 3 Hypothetical residence plots showing alternative explanations for a low number of recaptures.

The zero day data point is the number of specimens initially captured and released, and the day 1 data point is the number of specimens recaptured at least once; red dotted lines indicate the number of specimens never recaptured. If the residence-rate of the average phenotype is high (gradient of regression line shallow), specimens will remain in the population for a long time and a high number of recaptures is predicted (top line, diamonds). If the observed number of recaptures are low, there are two possibilities. (A) The residence-rate of the average phenotype has declined (gradient of regression line steep), so specimens rapidly exit the population and avoid recapture (bottom line, squares). (B) A second phenotype has appeared in the population with a very high death/emigration rate; these avoid recapture and do not impact on the calculated residence-rate of the average phenotype (gradient of regression line unchanged; bottom line, squares), but their presence can be deduced if the number of recaptures is significantly lower than would be predicted with the residence-rate of the average phenotype.

Method

Residence and recapture plots (from whose gradient the encounter-rate can be derived, see Appendix SI) were prepared from the MRR data for each year of the study period; best-fit lines were fitted using least-squares regression in Excel. The fraction of specimens predicted to be recaptured at least once (F) in each year was calculated from the formula (4) F=em/2gg+m

where m = gradient of best-fit line of residence plot and g = gradient of best-fit line of recapture plot. The derivation of this equation is given in Appendix SI.

The number of specimens predicted (or expected) to be recaptured at least once (ER) for any intra-seasonal period was calculated as N.F, where N is the number of specimens initially caught and released during that period. The standardized residual (ΔR) of the observed number of recaptures (OR) from the expected number was calculated as (5) ΔR=OR−ERER

If ΔR is positive/negative then more/less specimens were recaptured than predicted. This allows inter- and intra-seasonal changes in the proportion recaptured to be interpreted in terms of standardized departures from expectation.

Results

The male and female eclosion curves were not Gaussian; the pattern of emergences was usually complex and differed among seasons (Fig. 4). The male emergence curve was not truncated. The fitness curves (Fig. 4) derived from the whole-season residence-rates (Table 1) and emergence curves, under the assumption that all males have equal residence times, show that the number of matings a male can expect in the Dibbinsdale population varies with his eclosion time: males emerging early in the season are generally predicted to have higher fitness (≥1) than those emerging later (<1). In some years, males emerging ‘too early’ are also predicted to have low fitness, but this effect is intermittent. Hence, the null hypothesis that emergence timing does not affect male fitness, assuming all residence times are equal, can be rejected.

Figure 4 Predicted male fitness (A–C) in relation to male and female emergence curves (D–F) in the Dibbinsdale population for each year in the study period.

The expected fitness (number of matings, λ) for males emerging on each day of the season was calculated from Eq. (2); a fitness of 1 is marked by the horizontal solid line. Male (filled circles) and female (open circles) daily emergences were estimated with POPAN and are summed over 4-day periods. Vertical dotted lines show the sub-division of each year into separate intra-seasonal periods based on long-term changes in predicted fitness (usually to a value <1; the peak in late season 2006 has been neglected since it only corresponds to a small number of specimens; high fitness periods in early season 2006 and 2009 have also been distinguished).

Table 1 Summary of data obtained from residence and recapture plots in each year of the study period.

The slope of the best-fit line for the residence plot (m) was used to calculate the daily residence-rate (s) as exp(m) and the average residence time (RT) as −1/m. The fraction of specimens predicted to be recaptured at least once (F) was calculated from Eq. (4).

Year	m	S.E.	g	S.E.	s	RT	F	
2005	−0.169	0.006	−0.501	0.019	0.845	5.92	0.687	
2006	−0.374	0.020	−0.758	0.011	0.688	2.67	0.555	
2007	−0.289	0.014	−0.813	0.035	0.749	3.46	0.639	
2008	−0.152	0.008	−0.851	0.055	0.859	6.58	0.786	
2009	−0.162	0.006	−0.285	0.010	0.850	6.16	0.588	
2010	−0.207	0.009	−0.654	0.040	0.813	4.83	0.685	

When the emergence curve is partitioned into different intra-seasonal periods on the basis of long-term changes in the accompanying fitness curves (Fig. 4), it is found that a high percentage (44% on average) of males consistently emerge after their predicted fitness declines to an unsustainable level (<1) in late season (Table 2). This contrasts strongly with the isolated Durham population studied by Parker & Courtney (1983), in which the late season decline in fitness to a value <1 coincides with a sharp decline in male emergence frequency (Fig. 5) such that only 3% of the population emerged after this date (Table 2).

Figure 5 Male and female emergence curves and estimated male fitness (both summed over 4-day periods) for an isolated population of A. cardamines in Durham in 1977.

Note the sharp decline (truncation) in male emergence frequency after predicted fitness declines to <1 (vertical dotted line); in contrast, the female emergence curve is not truncated. (Reanalysed from data in Parker & Courtney, 1983.)

Table 2 Mean late season male fitness (λav, from Eq. (3)) against percentage emerging (from POPAN) in the Cheshire and Durham A. cardamines populations.

Population	Year	λav	% emerging	
Cheshire	2005	0.83	40	
	2006	0.68	48	
	2007	0.45	36	
	2008	0.61	61	
	2009	0.67	37	
	2010	0.46	41	
Durham	1977	0.80	3	

The presence of a high proportion of disadvantaged late emerging males in the Dibbinsdale population raises the question as to whether they remain within it (and hence whether the assumption that all males have equal residence times is valid). Males emerging in different intra-seasonal periods were recaptured at different rates. In all years, the fraction of males recaptured at least once declined through the season (Table 3). When the observed number of recaptures is compared with the expected number (on the hypothesis that all males belong to the same phenotype characterized by the whole-season residence-rates given in Table 1) it is found that in late season the number of recaptures always falls significantly below expectation (results of χ2 tests given in Table 3). In earlier intra-seasonal periods, the observed number of recaptures was usually close to the expected number, except in 2005, when it was significantly below expectation, and in 2006 and 2009, when in early season it was significantly above expectation.

Table 3 Variation in male average fitness (λav) and recapture frequency with intra-seasonal period (specified by long-term changes in daily fitness; see Fig. 4).

The results of χ2 tests on the significance of the departures from expectation are also shown.

Year	Period (d)	λav	N	OR	ER	χ 2	P	
2005	01–15	1.12	66	35	45.4	7.585	0.006	
	16–36	0.83	33	10	22.7	22.696	<0.001	
2006	01–04	2.23	20	18	11.1	9.630	0.002	
	05–08	0.90	41	22	22.8	0.057	0.811	
	09–32	0.68	43	10	23.9	18.121	<0.001	
2007	01–15	1.32	82	50	52.4	0.294	0.588	
	16–32	0.45	47	14	30.0	23.627	<0.001	
2008	01–09	1.47	33	22	25.9	2.813	0.094	
	10–25	0.61	39	12	30.7	53.180	<0.001	
2009	01–12	1.24	52	40	30.6	7.081	0.008	
	13–26	1.13	51	27	30.0	0.712	0.399	
	27–46	0.67	37	13	21.7	8.521	0.004	
2010	01–21	1.32	71	46	48.6	0.444	0.505	
	22–36	0.46	38	17	26.0	9.909	0.002	
Notes.

N number of males captured during the specified period

OR observed number of recaptures

ER expected number of recaptures (= NF, where F is taken from Table 1)

Multiple regression analysis showed that predicted male fitness but not year had a significant effect on the standardized residuals of the observed minus expected number of recaptures (ΔR); an apparent tendency for ΔR to increase over the years (when controlling for fitness) can therefore be neglected (p = 0.15). When year was removed from the model, the relationship between ΔR and predicted male fitness was strongly positive and highly significant (Fig. 6). If lower than expected recapture rates (negative ΔR) result from faster dispersal, these data support the hypothesis that males emigrate from the study area at times when their future mating prospects are low.

Figure 6 Regression of ΔR (standardized residuals from expected number of recaptures, from Eq. (5)) on male fitness (average number of matings, λav).

The vertical line marks a fitness of 1.0. Regression equation: ΔR = 2.81λav−3.99, p = 0.0004, R2 = 0.67. (Note that the two data points from early-season 2007 and early-season 2010 overlap (see Table 3); there are actually 14 data points contributing to the regression.)

In most seasons (2005–2009), reanalysis of late season data using the gradients of residence plots restricted to that period (e.g., 2009, Fig. 7) show that there were two co-occurring phenotypes in the population at that time. In those years, the late season deficit in the number of recaptures remained highly significant, even when the residence-rate of recaptured specimens decreased (Table 4). Therefore, even if the death/emigration rate of the average phenotype had changed in late season, it cannot wholly account for the low number of recaptures obtained during that period. Instead, this must be attributed to a very rapid loss of specimens from the study area shortly after first capture, and, by implication, shortly after emergence. While this might suggest that some butterflies undertake an escape flight, such behaviour has never been observed in the field, and it certainly does not occur in early season, or amongst recaptured specimens in late season. The excess specimens never recaptured must belong to a second phenotype characterized by a very high death or emigration-rate.

Figure 7 Intra-seasonal residence plots for 2009 and 2010.

The residence-rate is related to the gradient (m) of the plots; the steeper the slope, the faster specimens depart (die/emigrate) from the population and the lower the residence-rate. (A) In 2009, recaptured butterflies behaved uniformly through the season (m ± SE = −0.162 ± 0.011 for early (1–12 d) specimens; −0.162 ± 0.006 for mid (13–26 d) specimens; −0.187 ± 0.012 for late (27–46 d) specimens); therefore the low number of late season recaptures must be attributed to the appearance of a new phenotype which evaded recapture due to a high death/emigration rate (red dotted line). (B) In 2010, recaptured late season butterflies exited the population more quickly than early season ones (m ± SE = −0.182 ± 0.010 for early (1–21 d) specimens; −0.480 ± 0.041 for late (22–36 d) specimens); this was sufficient to account for the low number of recaptures during that period (red dotted line). In this case, the behaviour of the average phenotype had changed through the season. Compare Fig. 3.

Table 4 Reanalysis of late season recapture data using the gradients (m) of residence plots restricted to that period to recalculate the predicted fraction of recaptures (F).

For 2005–2009, the observed number of recaptures (OR) falls significantly below the expected number (ER), even when the late season values of m are steeper than the whole season values (shown in Table 1). For 2010, the late season value of m removes the recapture deficit obtained with the whole season value (Table 1).

Year	m	S.E.	F	N	OR	ER	χ 2	P	
2005	−0.163	0.017	0.695	33	10	22.9	23.919	<0.001	
2006	−0.523	0.040	0.456	43	10	19.6	8.622	0.003	
2007	−0.275	0.037	0.651	47	14	30.6	25.841	<0.001	
2008	−0.246	0.021	0.686	39	12	26.8	25.906	<0.001	
2009	−0.187	0.012	0.550	37	13	20.3	5.894	0.015	
2010	−0.480	0.041	0.454	38	17	17.2	0.006	0.938	

The only exception to this pattern occurred in 2010, when the low number of recaptures in late season can be wholly accounted for by a decreased residence-rate (Table 4); since late-season specimens had a shorter average residence time (2.1 days) than those emerging earlier (5.5 days), fewer were recaptured. In this case the average phenotype changed between early and late season, with the late season specimens exhibiting a higher death and/or emigration rate than the early season ones (Fig. 7).

The co-occurring late season phenotypes in 2005–2009 could differ in their death or emigration rates. To try to distinguish between these possibilities, we note that if the rapid loss of a high number of late season males were due to death, it would be unaffected by spatial scale, since a dead individual can never be recaptured no matter how wide the area in which we search. Therefore, estimates of the recapture shortfall (i.e., the expected minus observed number of recaptures) should not be biased higher or lower when calculated on the scale of the sub-sites of first-capture (SSFC) compared with the whole-site (WS). On the other hand, if the losses were due to dispersal, some specimens might be recovered in the WS after they have left their SSFC, so the recapture shortfall is predicted to be smaller in the WS.

The recapture shortfall is always smaller when calculated on the scale of the WS (Table 5); the chances of this happening in 5 successive years, when the estimates are expected to vary at random (i.e., when there is an equal chance that the calculated shortfall will be higher or lower in the WS), is 0.55 = 0.03. Over the 5 year period, about 20 specimens in the recapture shortfall for the SSFC were later recovered elsewhere in the WS. These losses were therefore due to dispersal rather than death. It is therefore likely that most of the remaining 70 specimens in the recapture shortfall had dispersed to the continuum (where they were unavailable for recapture).

Table 5 Late season recapture shortfall (expected minus observed number of recaptures) calculated from gradients of residence and recapture plots specific to sub-sites of first-capture (SSFC) and whole-site (WS).

Year	SSFC	WS	
2005	13.2	12.7	
2006	16.4	13.9	
2007	20.4	16.0	
2008	21.5	18.7	
2009	18.8	8.7	
Total	90.3	70.0	

Discussion

Interpretation of results

A simple model of male fitness in a core population of A. cardamines in northwest England shows that, under the assumption that all males have equal residence times, the number of matings a male can expect consistently declines to <1 in late season. Hence, the null hypothesis that (on the long average) males emerging at any time in the season can expect the same number of matings is falsified. The large proportion of males (∼44%) emerging in late season requires explanation.

In five of the six years in the study period, there were two sharply contrasting male phenotypes in late season. The residence-rate of the first of these did not differ, or differed relatively little, from that of the early season butterflies. The second phenotype was specific to late season and had a very low residence-rate, entailing a very high death or emigration-rate. We think the latter possibility far more likely.

Since only one of the two co-existing phenotypes in late season exhibits a low residence-rate, it is not the case that the whole population suffers an increased death/emigration rate at this time. This constrains the range of plausible explanations invoking an increased death-rate. In particular, increased mortality caused by deterioration in the physical environment or increased predation-rate can be ruled out, since all butterflies would be affected. Instead, we require a cause of mortality that would impact heavily on some butterflies but not on others. If some late season butterflies emerge from the chrysalis diseased, this would partition the population into infected/uninfected phenotypes exhibiting sharply contrasting death-rates, as required. However, we think this very unlikely: the disease would have to be present in a high percentage of males year after year; we have never encountered anything like it when rearing butterflies from wild larvae, and no specimens caught in the field have ever shown any obvious signs of ill-health.

On the other hand, the occurrence of two sharply contrasting dispersal phenotypes in late season presents no such difficulties. Dispersal polymorphisms are common in nature. Moreover, the appearance of a migratory phenotype in our study population in late season is predicted by decreased mate availability at that time; lack of mates is a well established dispersal cue among many taxa. It is therefore likely that the newly appearing phenotype in late season is characterized by a very high emigration-rate; this interpretation is supported by the larger recapture shortfall obtained on a smaller spatial scale within the study site (Table 5), implying that some migratory specimens were recovered when the search area was widened.

In 2010, there was only one late season phenotype. This was characterized by a lower residence-rate than the early season butterflies. While an increase in the general death-rate in late season cannot be ruled out in this case, in view of the results obtained in all other years it seems likely that this too was at least partially an effect of rapid dispersal. Perhaps the migratory response was slower in this year, allowing a significant number of dispersing specimens to be recaptured and hence affect the calculated residence-rate.

We suggest that the appearance of a migratory phenotype in late season is part of a ‘stay-or-go’ response, in which males respond to cues relating to the availability of females by either settling in the Reserve (‘stay’ response) or rapidly emigrating from it (‘go’ response). Hence, the assumption of equal residence times in our fitness model is violated. Therefore, the model does not capture the true fitness of late emerging males; rather it reveals what is avoided by the existence of the ‘stay-or-go’ response, providing an insight into its evolution and/or maintenance. Were the ‘stay-or-go’ response to fail, then late season males should be selected against, since their fitness would be consistently lower than early season ones.

The prediction that spatial variation in population density can impact on the evolution of emergence timing and dispersal in heterogeneous landscapes (Fig. 2) is therefore supported by our data. In particular, we have evidence that selection for a truncated male emergence curve has been weakened in a high density core population by the evolution of a ‘stay-or-go’ response, which enables disadvantaged late emerging males to recover fitness by emigrating to a low density continuum. In contrast, the male emergence curve is truncated (or nearly so) in the isolated population studied by Parker & Courtney (1983), where a ‘stay-or-go’ response would be ineffective.

Evolution of emergence timing and dispersal in heterogeneous landscapes

Late emerging core males which emigrate to the continuum will not necessarily achieve the same fitness as earlier emerging ones, whose reproductive success is predicted to be >1. Instead, they behave in the same way as subordinates do in populations exhibiting source–sink structure (Pulliam, 1988), by moving from the core (source) habitat to a nearby (sink) area to increase their reproductive success above the level achievable in the core (they effectively “make the best of a bad job” (Maynard Smith, 1982)). Pulliam (1988) showed that such a strategy would be evolutionarily stable, and could result in the maintenance of sink populations which would otherwise go to extinction.

The number of matings obtained by late emerging core males which migrate to the continuum will depend on the trade-off between improved input timing and decreased mate encounter-rates. If the population in the continuum is too sparse, then they will not improve their fitness by emigrating to it, and so late emergence should be selected against, driving the male emergence curve as far as possible towards an ESS. This will depend on how far the shape of the emergence curve can depart from Gaussian (Bulmer, 1983), and, in stochastic environments, on the accuracy of emergence cues (Iwasa & Haccou, 1994).

If mate encounter-rates in the continuum are high enough, late emerging core males will improve their fitness by migrating to it, since protandry is less favoured in low density populations (Zonneveld & Metz, 1991). The evolution of an ESS emergence schedule would then be prevented by selection for late season dispersal. Since the emigration of competitors will reduce mate competition in the core, the fitness of males which stay behind will also increase. Hence, frequency-dependent selection likely explains the evolution of the ‘stay-or-go’ response observed in the Dibbinsdale population of A. cardamines. The long term maintenance of late emerging males in core populations will ultimately depend on the descendants of the ‘go’ individuals returning to them; although reproductively successful ‘stay’ males eclosing near the end of the season could perpetuate late emergence short term, one way loss of genes triggering the ‘go’ response would eventually lead to the intense level of scramble competition predicted to select against late eclosion.

The ‘stay-or-go’ response in the Dibbinsdale population of A. cardamines most likely results from a genotype-by-environment interaction, since it is strongly associated with predicted fitness (Fig. 6), suggesting that males are capable of adjusting their behaviour in response to environmental cues (“condition-dependent” or “informed” dispersal in recent literature (e.g., Clobert et al., 2009; Chaput-Bardy et al., 2010; Hovestadt, Mitesser & Poethke, 2014). In this connection, we note that males were sometimes (2006 and 2009) recaptured in excess in high fitness windows in early season (Table 3), indicating that they were more philopatric than normal when the environment was particularly favourable. Nevertheless, genetic variability in the reaction norms underlying the response likely explains why, in late season, some males ‘stay’ while others ‘go.’

Implications for conservation: maintenance of sink populations

A sink p+opulation is one which cannot persist without immigration, since the average reproductive success within it is <1 (Pulliam, 1988). We have shown that the emigration of disadvantaged late emerging male orange-tips from the core population in Dibbinsdale resembles the behaviour of subordinates in a source. If the continuum is a sink, the influx of subordinate males could play an important role in its persistence by boosting the number of females emerging there which are mated. In Appendix SII, we show how variance in male fitness (σ2), assumed to be generated by imperfect emergence timing, in a source population of size n is related to the size (nsk) of the population which can be maintained in a sink, on the assumption that disadvantaged males emerging in the source emigrate to the sink. This leads to the following equation: (6) Φλg−1σ=nskα−λgnλg

where λg is the average number of matings males can expect in the sink, α is the proportion of sink females mated, and Φ is the cumulative probability function of the normal distribution, which in this case gives the proportion of specimens emerging in the source with a lower predicted fitness than they would (on average) achieve in the sink, i.e., with fitness <λg. The size of the sink population can be obtained from (7) nsk=n1−1λsλgα−λg

where λs is the average number of matings obtained by ‘stay’ males in the source. These equations show how the variance in predicted male fitness in the source can be maintained by the existence of a sink population; or, conversely, how the maintenance of the sink population may be dependent on this variance.

It is instructive to quantify the relationship between precision in emergence timing and the potential size of a sink population in the Dibbinsdale area, where the size of the source population is n ≈ 300 (POPAN estimate). For simplicity, we assume all sink females are mated (α = 1). The average predicted fitness of source males emerging in early/mid season is 1.33 (Table 3); since our model neglects the emigration of ‘go’ specimens, the fitness of the ‘stay’ ones will be underestimated, so we set λs = 1.40. The average predicted fitness in the source in late season is 0.62 (Table 3); to make emigration to the sink viable, we set λg = 0.75. With these values, nsk ≈ 260 (Eq. (7)) and Φ = 0.29 (Eq. (6)), so σ2 = 0.20 (from tables). Therefore, a sink population of 260 can sustain imprecise emergence timing in the source which translates into a fitness variance of 0.20; conversely, if σ2 = 0.20 in the source, 29% of males will emigrate to the sink, where they will help to maintain a stable population of 260.

This approximation demonstrates the extent to which landscape structure could impact on the evolution of emergence timing, and so offers an explanation as to why the predictions of protandry theory, based on the implicit assumption of a homogeneous landscape, may fail. It also indicates that male behaviour could be very important for the maintenance of sink populations. In the original model of Pulliam (1988), sink populations are maintained by successive (partial) repopulations by subordinates emigrating from sources; in the absence of these repopulations, the low reproductive success (<1) of specimens in the sink would lead to their eventual extinction there. While this process (which in insects would depend upon the movement of egg-laying source females) may contribute to the maintenance of A. cardamines sink populations, our model indicates that an alternative mechanism may be equally important. We assume that Allee effects (decreased population growth with decreased population density due to reduced mate encounter rates) in sink populations are prevented by the arrival of subordinate males (determined by emergence timing) from source populations. Hence, the sink population is not rescued by the repeated input of subordinate source females, but rather maintained by the redistribution of subordinate source males. The implications for conservation are clear: even if females appear to be adequately adapted to find widely scattered oviposition sites in a sink, their reproductive success may depend on males arriving from a source; if the source population goes to extinction, then so will the population in the sink.

Supplemental Information

Appendix SI Derivation of predicted fraction of recaptures, F

Click here for additional data file.

Appendix SII Variance in source emergence timing and maintenance of sink populations

Click here for additional data file.

Figure S1 Histogram showing the distribution in predicted male fitness values over the six-year study period (2005–2010), which has a standard deviation of 0.46

The Gaussian distribution with μ = 1 and σ = 0.46 (filled circles) is provided for comparison.

Click here for additional data file.

We thank the Rangers at Dibbinsdale, Peter Miller and Alan Smail, for permission to undertake the work in the Reserve. Jenny Hodgson, Stephen Cornell and two anonymous reviewers provided valuable comments on the manuscript.

Additional Information and Declarations

Competing Interests

Author Contributions

The authors declare there are no competing interests.

W James Davies conceived and designed the experiments, performed the experiments, analyzed the data, wrote the paper, prepared figures and/or tables.

Ilik J. Saccheri reviewed drafts of the paper.

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
