# Peer review of "Male emergence schedule and dispersal behaviour are modified by mate availability in heterogeneous landscapes: evidence from the orange-tip butterfly"

_PeerJ, doi:10.7717/peerj.707_

## Round 0.1 · original submission · Major Revisions

Please make a revised version together with a detailed list of replies to each comment by the reviewers. The revised paper will be sent back to the same two reviewers in this round.

Reviewer 1 ·

Basic reporting

This manuscript argues that on the basis of mark-and-recapture method, male emergence schedule in orange-tip butterflies, Anthocharis cardamines, living in the core habitat (‘source’) surrounded by more thinly populated area (‘sinks’) in heterogeneous landscapes, is not consistent with that predicted by a protandry model, particularly with regard to an isolated population, and the discrepancy should be related to the late emerging males that partially disperse to the sink area. The authors then discuss special behavioral characteristics of late emerging males (‘stay’ or ‘go’) leading to the dispersal of these males, stochastic accuracy of emergence timing in the source population and application of their idea to conservation in a heterogeneous landscape. Although the authors’ arguments and discussions are very interesting and important from the viewpoint of insect population ecology in a heterogeneous landscape, the most important problem is that the evidence and analysis which the authors proposed to support their argument, particularly about whether decreased recapture rate late in the season is caused by dispersal or death, is insufficient to support their claims. Fundamentally, it is impossible to discriminate whether the disappearance of individuals is caused by death or emigration based solely on the present mark-and-recapture data for the core population. The estimates for emergence schedule suffer from the same problem. To clarify the real pattern of emergence and dispersal, the authors need additional information from a mark-and-recapture study including both the source and sink area. At the present stage, therefore, the authors should reconstruct their conclusions and discussion including the possibility that the disappearance of individuals could be caused by death as well as dispersal to surrounding areas. Otherwise, authors need migration data between the source area and the surrounding sink areas.

Experimental design

If the authors want to examine estimates of residence-rate, recapture-rate and fitness in order to clarify intra-seasonal changes in any internal parameters of the core population, then the present methods are appropriate. However, if they want to elucidate any other parameters relating the core population to the sink population, e.g. dispersal-rate, then they must perform the mark-and-recapture including both core area and sink area.

Specific problems as follows:

Equation (5): The denominator may be wrong. The correct one should be
j=Z
Σ Mj
j=y
L220-227: The authors neglected male and female immigrants into the core population because of the relatively higher density when estimating the curve of emergence. But the authors must show evidence according to this treatment to argue the validity of this assumption.
L231: “matable” ?
L251: Put “estimates” instead of “data”
L164: Do the authors have any evidence that male behaviour is influenced by size and timing of emergence?
L164: Are there any groups with different size class and the same intra-seasonal period?

Validity of the findings

L272-276, Figure 3: This plot has a problem of independence for data units, especially not separating between inter- and intra-seasonal effects. Deviations of each estimate (ΔR and male fitness) from the mean of each year should be plotted to examine their relationship.
L277-290: It is extremely difficult to understand what the authors are attempting to explain about death and emigration in individuals. Any effects of death and dispersal to MRR procedure are exactly the same. The activity of live individuals and several factors causing a shortened life span can probably induce various patterns in residence-rate and the number of recaptures. Fundamentally, it is impossible to discriminate death and emigration as cause for lost individuals from the present data examining the core population only. If the authors argue that negative ΔR results from dispersal rather than death, they need to present evidence, e.g. seasonal change of the number of marked males outside the core area.
L291-298: These arguments are too speculative as a content of “Results”, because there is no evidence.
Figure2: What is the scale unit on one of the vertical axis, ‘Population Emerging’ ?
L343: There is a possibility of movement for “ideal free distribution”.
L287-298: Did you obtain any evidence for ‘stay-or-go’ response in males’ behavior during the present study? Without this type of data, you had better move this to the ‘Discussion’ part. Additionally, the authors should compare residence-rate, recapture rate and the intra-seasonal difference of these estimates in males with those in females to examine whether any male-specific characteristics according to male mating strategy will be found.
L378: put “inferred” instead of “uncovered”
L833-837: Unfortunately, only the appearance of normal distribution from sampling of fitness estimates is not sufficient proof of your assumption, because the normal distribution is just error distribution, so with such large samplings even from variously formed distribution of population will inevitably approximately generate a normal distribution. That is a statistical principle. Probably, similar sampling from the data of Parker and Courtney (1983) would result in a normal distribution. Authors had better remove Appendix III because it is not so important.

Additional comments

Generally, the authors could not discriminate emergence and immigration, although they determined that the main entrants should be from emergence in the core population, nor could they discriminate between death and emigration in spite of their conclusion that the decrease in the number of recaptures late in the season should be from emigration. Those are the most important facts on which their following discussion is based. Therefore, the authors need additional information from the mark-and-recapture study in both the core population and the surrounding area in order to justify their discussion, otherwise it remains as speculation.

Reviewer 2 ·

Basic reporting

The article included a comprehensive background that gave a detailed history of the topic. But it was easy to get lost in all the theories and different predictions which were contained in very condensed and therefore dense prose. I would have found a figure, and possibly a table, to sort out the different theories, predictions and what had already been solidly supported by data. By the time I got to the description of what they were going to test and why, it was hard for me to judge how novel and interesting it was. It wasn't until I really got to the discussion that I understood that the primary test (does male fitness vary across season, which to me was fairly settled) was really a pretext to determine if declining male abundance was actually due to protandry or males fleeing to find females in marginal habitat or populations.

Experimental design

It was very difficult for me to see through a lot of mark-recapture jargon to find the basic design of the experiment that allowed them to test their idea. If I understand correctly (and I may not), their idea is that isolated populations are less likely to show male "stay or go" behavior compared to populations that are embedded in a matrix of a low-density sink population. Therefore, "escape" behavior should be more common in less isolated populations because males could improve fitness by leaving and finding females, no matter how scarce. My understanding is that they do this by reanalyzing data in an isolated population from a study of a single population from one year (Durham 1977, as reported in Parker and Courtney 1983) and then comparing it to their population in Dibbinsdale where they had 5 years of data (2005-2010). If I read this correctly, there are two problems. It doesn't seem fair to have your two main populations separated by so much space and time. Certainly, there are many other differences between them. Further, there is no replication for the "isolation" treatment (1 population in one year) and pseudo replication for the "source-sink" treatment (1 population across 5 years). Finally, it was hard for me to see in the results how the populations really differed.

Validity of the findings

In addition to the lack of replication for the treatments, the conclusions of the study really seem to be dependent on believing that recapture rates can really be understood to be reliable estimates of emigration and not death. As someone who is familiar with MR methods, but not steeped in them - I found it very heard to follow a lot of jargon about assumptions, etc. I think a stronger foundation to help me have confidence that recapture rates in this setting can really be ascribed to emigration rather than other causes. Further, it seems to me that the conclusions would be much stronger if you had some additional evidence. Two types jump instantly to mind. One is some behavioral studies that would estimate emigration rates based on tracking studies and following individuals near the boundaries of habitats (where you can observe them staying or going). The second would be some evidence that emigrating males are finding females in the low-density matrix and improving their fitness. Otherwise, just the differing estimates of 'dispersal' (based on a difference in recapture rates) seemed kind of weak.

Additional comments

I thought this was a very interesting topic - but it was hard for me to follow the initial rationale for the study. It is quite possible that some clarifying language would clear up some misconceptions I had about your methods and conclusions.

---

## Round 0.2 · accepted · Accept

Your revision has been assessed by one reviewer and he has found it adequate. My own reading of the paper supports this assessment.

Reviewer 1 ·

Basic reporting

No Comments

Experimental design

No Comments

Validity of the findings

No Comments

Additional comments

No Comments